# Macropinocytosis drives T cell growth by sustaining the activation of mTORC1

John C. Charpentier [1], Di Chen[1], Philip E. Lapinski[1], Jackson Turner[1], Irina Grigorova[1], Joel A. Swanson [1] & Philip D. King[1]*

Macropinocytosis is an evolutionarily-conserved, large-scale, fluid-phase form of endocytosis that has been ascribed different functions including antigen presentation in macrophages and dendritic cells, regulation of receptor density in neurons, and regulation of tumor growth under nutrient-limiting conditions. However, whether macropinocytosis regulates the expansion of non-transformed mammalian cells is unknown. Here we show that primary mouse and human T cells engage in macropinocytosis that increases in magnitude upon T cell activation to support T cell growth even under amino acid (AA) replete conditions. Mechanistically, macropinocytosis in T cells provides access of extracellular AA to an endolysosomal compartment to sustain activation of the mechanistic target of rapamycin complex 1 (mTORC1) that promotes T cell growth. Our results thus implicate a function of macropinocytosis in mammalian cell growth beyond Ras-transformed tumor cells via sustained mTORC1 activation.

---

[1] Department of Microbiology and Immunology, University of Michigan Medical School, Ann Arbor, Michigan 48109, USA. *email: kingp@umich.edu

Macropinocytosis is an evolutionarily conserved, large-scale, fluid-phase form of endocytosis observed in single-celled organisms, such as *Dictyostelium*, as well as select cell types in multicellular organisms[1–3]. In *Dictyostelium*, macropinocytosis provides nutrients for cell growth. In mammals, macropinocytosis has been ascribed different functions including antigen presentation in macrophages and dendritic cells and regulation of receptor density in neurons[4–6]. More recently, macropinocytosis has been shown to regulate the growth of Ras-transformed tumor cells under conditions where the availability of extracellular amino acids (AA) is limiting[7–9]. Under such conditions, proteins taken up by macropinocytosis are degraded into AA, which fuel cell growth and survival by incorporation into central carbon metabolism and by activation of the mechanistic target of rapamycin complex 1 (mTORC1). However, whether macropinocytosis regulates the growth of non-transformed mammalian cells is unknown.

To determine if macropinocytosis is required for the growth of non-transformed cells, we ask if the growth of primary T lymphocytes is dependent upon macropinocytosis. T cells are an appropriate cell type with which to address this question since upon activation they enter the cell cycle rapidly, show substantial increases in cell size during the G1 phase of the cell cycle, and have high proliferative capacity. Here we report that macropinocytosis is necessary for T cell antigen receptor (TCR)/CD28-induced growth of T cells even when AA are available in the extracellular space. Macropinocytosis in activated T cells provides a means of access of extracellular AA to an endolysosomal compartment necessary for sustained activation of mTORC1 that drives T cell growth. Thus, a role for macropinocytosis in mammalian cell growth is not limited to Ras-transformed tumor cells but may be a more general phenomenon involved in the growth of non-transformed cells.

## Results

**T cells engage in macropinocytosis.** Macropinocytosis has not been reported previously in T cells. To determine if T cells engage in macropinocytosis, we tested the ability of murine splenic T cells to endocytose 70 kDa fluorescein-dextran (Fdex) that is used commonly as a macropinocytosis probe[3,10,11]. The size of 70 kDa Fdex limits its uptake into cells by small-scale endocytic processes such as clathrin-mediated endocytosis[11,12]. As assessed by flow cytometry, murine CD4+ and CD8+ T cells readily took up 70 kDa Fdex (Fig. 1a–c and Supplementary Fig. 1a, b). The extent of probe uptake increased upon activation of T cells with monoclonal antibodies (mAb) against the CD3 component of the TCR complex and the T cell co-stimulatory receptor, CD28, coincident with an increase in cell size. T cell fluorescence upon probe incubation was essentially abolished at 4 °C (Fig. 1a and Supplementary Fig. 1a). Thus, acquisition of T cell fluorescence is a consequence of an energy-dependent endocytic process and not binding to a cell surface receptor.

In order to visualize putative T cell macropinosomes, we used fluorochrome-labeled bovine serum albumin (BSA) as an alternative probe. Similar to 70 kDa Fdex, BSA is taken up into cells primarily by macropinocytosis[7,9]. However, in contrast to 70 Fdex, BSA is retained in cells after fixation. Flow cytometric analyses confirmed uptake of fluorochrome-labeled BSA by murine CD4+ and CD8+ T cells that increased upon CD3/28 mAb stimulation coincident with an increase in T cell size (Fig. 1d–f and Supplementary Fig. 1c, d). As shown by confocal microscopy of fixed cells, labeled BSA was identified within distinct vesicles of CD3/28 mAb-stimulated T cells that ranged in number from 1 to 10 or more per cell (Fig. 1g, h and Supplementary Fig. 1e). The size of vesicles is consistent with

their identity as macropinosomes that range in size from 200 nm to 1 µm in diameter[1–3].

To determine if T cell endocytosis of BSA increased upon TCR recognition of cognate peptide-MHC ligands, we used T cells from OTII TCR transgenic (Tg) mice that are specific for ovalbumin (Ova) peptide 323–29 in complex with the I-A$^b$, MHC class II molecule[13]. Upon adoptive transfer to wild-type mice, OTII TCR Tg T cells showed increased uptake of BSA in response to immunization of recipients with Ova protein (Fig. 1i–k). Human peripheral blood CD4+ and CD8+ T cells also took up BSA in a temperature-dependent fashion and the extent of uptake was increased upon stimulation with CD3/28 mAb (Fig. 1l–n) or the T cell mitogen phytohemagglutinin (PHA) (Supplementary Fig. 1f, g). Thus, endocytosis of macropinocytosis probes is not unique to murine T cells.

We used scanning electron microscopy (SEM) to identify forming macropinosomes in murine T cells. Macropinosomes at different stages of development were readily identified in both unstimulated and CD3/28-stimulated T cells, although were more frequently observed in the latter, coincident with increased cell surface ruffling (Fig. 2a, b). Macropinocytic structures in T cells strongly resembled those observed in CSF1-stimulated murine bone marrow (BM) macrophages (Fig. 2c). Macropinocytic cup formation is driven by actin polymerization, in contrast to most small-scale forms of endocytosis[1–3,11]. Consistent with the identity of T cell surface structures identified by SEM as developing macropinocytic cups, staining of CD3/28-stimulated T cells with phalloidin revealed polymerized actin in loops that extended 1–2 µm from the cell surface (Fig. 2d).

To establish that T cell uptake of large molecular weight probes is mediated by macropinocytosis, we examined the effect of 5-(N-Ethyl-N-isopropyl) amiloride (EIPA), which is a highly specific inhibitor of macropinocytosis[1–3]. EIPA blocks the function of a plasma membrane Na+/H+ exchanger that is required for macropinocytosis[14]. We also examined the influence of the inhibitors of actin cytoskeletal dynamics, jasplakinolide in combination with blebbistatin (J/B)[15]. EIPA and J/B inhibited uptake of BSA by CD3/28 mAb-stimulated murine and human CD4+ and CD8+ T cells at concentrations previously reported to block macropinocytosis in other cell types (Fig. 3a–h and Supplementary Fig. 2g). The same results were obtained with purified murine T cells and when 70 kDa Fdex was used as a probe (Supplementary Fig. 2a, b). In contrast, PitStop, an inhibitor of clathrin-mediated endocytosis[16], had no influence upon probe uptake (Fig. 3b and Supplementary Fig. 2a, b). Uptake of BSA by unstimulated murine T cells was also blocked by EIPA and J/B but not PitStop showing that constitutive probe uptake is also mediated by macropinocytosis (Supplementary Fig. 2f).

**Mechanism of T cell macropinocytosis.** Macropinocytosis is dependent upon activation of the Ras small GTPase in most cell types[15,17,18]. In classical macropinocytosis, active Ras triggers activation of phosphatidylinositol 3-kinase (PI3K) leading to activation of Rac1 and Cdc42 Rho-family GTPases that orchestrate actin polymerization and macropinocytic cup formation[1–3]. The serine/threonine protein tyrosine kinase Pak-1 has also been implicated in cup formation and closure[1–3]. To further characterize T cell macropinocytosis, we tested the effect of inhibitors of these signaling molecules. Farnesyl thiosalicylic acid (FTS), which rapidly displaces H-, N-, and K-Ras isoforms from membranes resulting in their rapid degradation[19], had no effect upon probe uptake by CD3/28 mAb-stimulated T cells (Fig. 3b). In contrast, LY294002[20], EHT 1864[21] and IPA-3[22], which inhibit PI3K, Rac1 and Pak1, respectively, partially blocked probe uptake

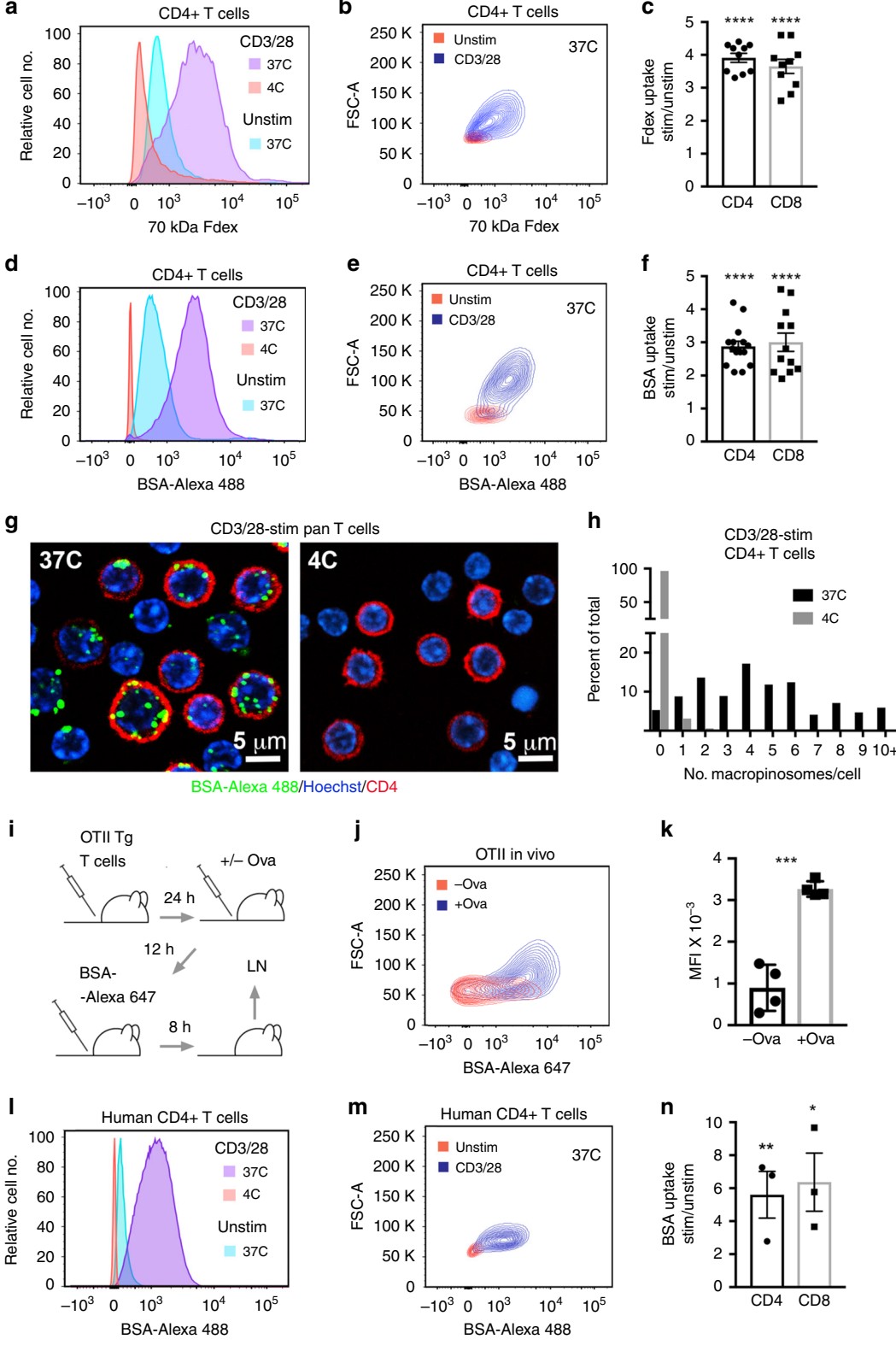

(Supplementary Fig. 2c). We also examined macropinocytosis in T cells from mice that lack the RasGRP1 guanine nucleotide exchange factor, which is required for Ras activation in this cell type[23]. Loss of RasGRP1 in T cells did not affect T cell uptake of BSA (Supplementary Fig. 2d, e). Thus, T cell macropinocytosis differs from classical macropinocytosis in that it is independent of Ras, although shows partial dependency upon PI3K, Rac1, and Pak1. Similar results were obtained with unstimulated murine

T cells with the exception that LY294002 had no inhibitory effect (Supplementary Fig. 2f).

**T cell macropinocytosis is necessary for G1 growth**. T cells activated with CD3/28 mAb show major increases in cell size between 12 and 20 h post-stimulation, which corresponds to the G1 phase of the cell cycle[24] (Supplementary Fig. 3a). Therefore, to determine if macropinocytosis is necessary for G1 growth, we tested

**Fig. 1 T cell uptake of macropinocytosis probes. a–f** Murine splenocytes (**a–f**) were unstimulated or stimulated with CD3/28 mAb for 24 h (**a–c**) or 20 h (**d–f**). 70 kDa Fdex (**a–c**) or BSA-Alexa 488 (**d–f**) probes were added to cells for the last 4 h or 8 h of culture, respectively, at the indicated temperatures. **a, b, d, e** Representative flow cytometry histogram plots of CD4+ T cell probe uptake and contour plots of probe uptake vs. FSC-A. **c, f** Mean ± 1 SEM of the ratio of Fdex (**c**) or BSA (**f**) probe uptake in stimulated vs. unstimulated CD4+ and CD8+ T cells at 37 °C (**a–c**, n = 10 independent experiments; d–f, n = 15 and 12 independent experiments for CD4+ and CD8+ T cells, respectively). ****P < 0.0001 by Student's 1-sample 2-sided t-test. **g** Images show temperature-dependent uptake of BSA-Alexa 488 by CD3/28 mAb-stimulated purified murine pan T cells (probe incubation from 12 to 20 h) into structures that resemble macropinosomes. Representative images of six repeat experiments are shown. **h** Quantitation of the number of macropinosomes per CD4 T cell (n = 169 and 196 cells at 37 °C and 4 °C, respectively). **i** Method used to assess uptake of BSA-Alexa 647 by OTII TCR Tg T cells in vivo following immunization with whole Ova protein. **j, k** Representative flow cytometry contour plot of probe uptake vs. FSC-A (**j**) and mean ± 1 SEM of median fluorescence intensity (MFI) of Alexa 647 fluorescence of OTII TCR Tg T cells from unimmunized and immunized mice (n = 4 mice for each condition) (**k**). ***P < 0.001 by Student's 2-sample 2-sided t-test. **l–n** Human PBMC were unstimulated or stimulated with CD3/28 mAb for 20 h. BSA-Alexa 488 was added to cells for the last 8 h of culture at the indicated temperatures. **l, m** Representative flow cytometry histogram plots of CD4+ T cell probe uptake and contour plots of probe uptake vs. FSC-A. **n** Mean ± 1 SEM of the ratio of BSA uptake in stimulated vs. unstimulated CD4+ and CD8+ T cells at 37 °C (n = 3 independent experiments). *P < 0.05, **P < 0.01, by Student's 1-sample 2-sided t-test. Source data are provided as Source Data file.

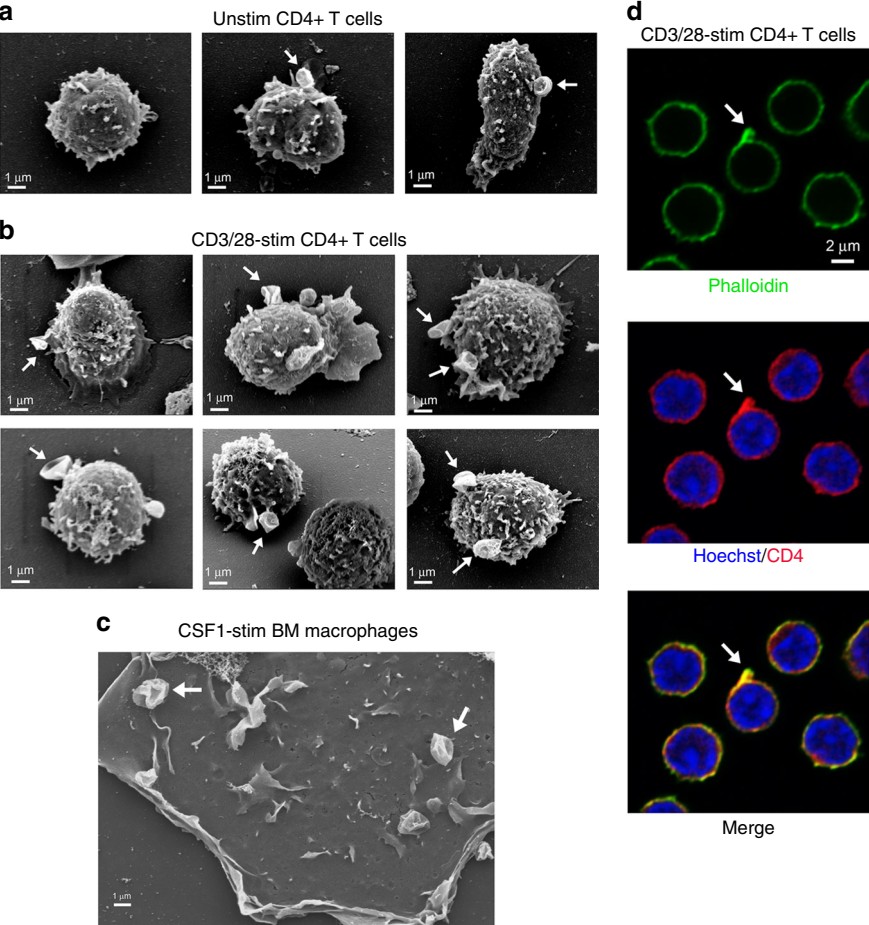

**Fig. 2 T cell macropinocytosis. a, b** SEM images of murine CD4+ T cells, unstimulated or stimulated with CD3/28 mAb for 16 h. Macropinocytic cups at different stages of development are indicated (arrows). **c** SEM image of a murine BM macrophage stimulated with CSF1 for 15 min. Note similarity of macropinocytic cups to those identified in T cells (arrows). **d** Murine CD4+ T cells, stimulated with CD3/28 mAb for 16 h, were fixed and permeabilized, stained with Alexa 488-labeled phalloidin and anti-CD4 mAb, and analyzed by confocal microscopy. Shown images are 3 μm above the plane of T cell contact with the substratum. Note cell surface projected loop of polymerized actin (arrows).

the effect of inhibitors of macropinocytosis upon growth when added to cultures immediately prior to this period. EIPA and J/B were potent inhibitors of CD4+ and CD8+ T cell growth (Fig. 4a, Supplementary Figs. 3b, 4a, b). In addition, other macropinocytosis inhibitors restricted T cell growth in these experiments (Fig. 4a and Supplementary Fig. 3b). Overall, we observed a strong correlation between the influence of an inhibitor upon macropinocytosis and T cell growth (Fig. 4b and Supplementary Fig. 3c). Importantly, at the concentrations used, inhibitors showed minimal toxicity during the

12–20 h incubation period. These findings support the contention that macropinocytosis is required for T cell growth even under conditions of optimal nutrient availability.

**Macropinocytosis-dependent mTORC1 activation in T cells.** In AA-starved Ras-transformed tumor cells, macropinocytosis has been linked to activation of mTORC1[8,9]. Endolysosomal AA derived from macropinocytosed and subsequently digested

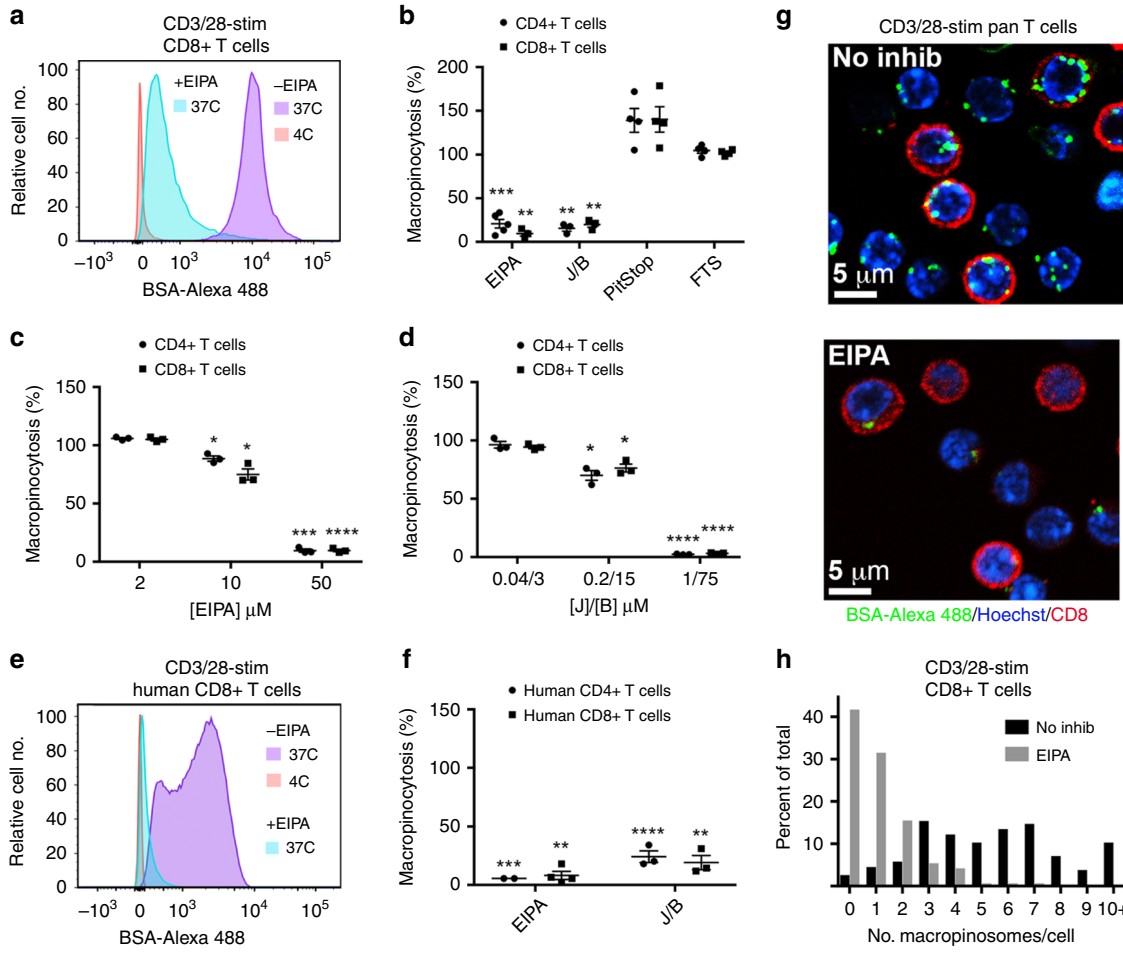

**Fig. 3 Inhibitors of macropinocytosis block T cell uptake of BSA. a–f** Murine splenocytes (**a–d**) or human PBMC (**e**, **f**) were stimulated with CD3/28 mAb for 12 h before incubation with BSA-Alexa 488 at 37 °C or 4 °C in the presence or absence of the indicated inhibitors for a further 8 h. In (**a**, **b**, **e**, **f**), EIPA and J/B were used at the highest concentrations shown in (**c**) and (**d**), respectively. **a**, **e** Representative flow cytometry plots showing the influence of EIPA upon probe uptake by CD8+ T cells. **b–d**, **f**, Mean ±1 SEM of the percentage macropinocytosis relative to the positive control, calculated as indicated in Methods. **b** (EIPA, $n = 5$ for CD4+ and $n = 3$ for CD8+; J/B, $n = 3$; Pitstop, $n = 4$; FTS, $n = 4$ independent experiments). **c**, **d** $n = 3$ independent experiments. **f** $n = 2$ and 4 independent experiments for CD4+ and CD8+, respectively, with EIPA and $n = 3$ independent experiments for J/B. *$P < 0.05$, **$P < 0.01$, ***$P < 0.001$, ****$P < 0.0001$ by Student's 1-sample 2-sided $t$-test. **g** BSA-Alexa 488 uptake by CD3/28 mAb-stimulated purified murine T cells (probe incubation from 12 to 20 h) in the absence and presence of EIPA at 37 °C. Images are representative of five repeat experiments. **h** Quantitation of the number of macropinosomes per CD8+ T cell in the absence and presence of EIPA ($n = 156$ and 168 cells, respectively). Source data are provided as Source Data file.

proteins in these cells are thought to provide one of two signals required for activation of mTORC1 at the lysosomal surface[8,9,25,26]. Although in AA-starved Ras-transformed tumor cells mTORC1 activation restricts growth, under AA replete conditions, mTORC1, through phosphorylation of ribosomal protein p70 S6 kinase and other targets, promotes anabolic processes that drive growth[25,26]. Therefore, we examined the possibility that macropinocytosis is required for T cell growth because it is necessary for delivery of AA to an endolysosomal compartment at which site they promote mTORC1 activation. With the use of DQ Red BSA that exhibits fluorescence only upon digestion in lysosomes[27], we first confirmed that macropinocytosed material in activated T cells is targeted to lysosomes (Fig. 4c, d and Supplementary Fig. 3d, e). Subsequently, by flow cytometric staining for phospho-S6, we established that mTORC1 remains active in CD4+ and CD8+ T cells at 12 and 20 h after CD3/28 mAb stimulation (Fig. 4e and Supplementary Fig. 3g). Addition of Torin1, a potent inhibitor of mTORC1[28], to T cell cultures at 12 h abolished phospho-S6 staining at 20 h, confirming that phospho-S6 staining reflects ongoing mTORC1 activation during

this period (Supplementary Fig. 3f, h). Next, to determine if macropinocytosis is necessary for sustained mTORC1 activation between 12 and 20 h, the effect of addition of EIPA and J/B at 12 h upon mTORC1 activity was examined. EIPA and J/B (but not PitStop) blocked mTORC1 activation in CD4+ and CD8+ T cells in these experiments; an effect that was apparent as soon as 1 h after addition of the drugs (Fig. 4e and Supplementary Figs. 3g, h and 4c, d). In contrast to their effects upon mTORC1 activation, EIPA and J/B did not block sustained activation of the NFκB transcription factor in CD4+ or CD8+ T cells during the same period (Supplementary Fig. 4e–h). Furthermore, EIPA inhibited activation of mTORC1 in T cells that were stimulated acutely with CD3/28 mAb but did not inhibit activation of ERK mitogen-activated protein kinases (MAPK) or NFκB in the same short time course stimulation experiments (Supplementary Fig. 5a–e). Importantly, Torin 1 inhibited growth of CD4+ and CD8+ T cells when added to cultures at 12 h, albeit less potently than EIPA and J/B (Fig. 4a and Supplementary Fig. 3b). Thus, a role for macropinocytosis in T cell growth can be attributed, at least in part, to a function in sustained activation of mTORC1.

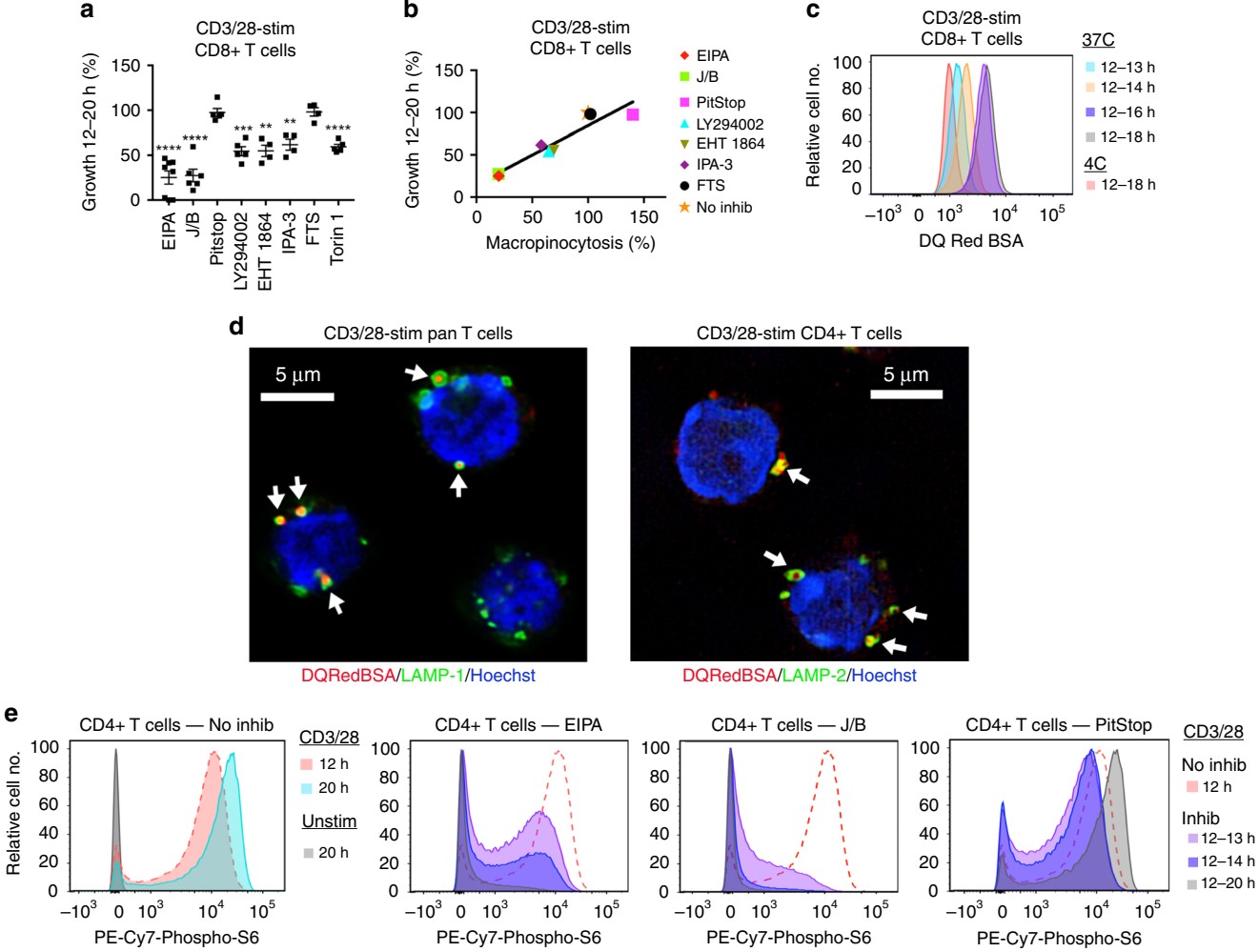

**Fig. 4 Macropinocytosis promotes T cell growth by sustaining the activation of mTORC1. a** Murine splenocytes were stimulated with CD3/28 mAb for 20 h in the absence or presence of the indicated inhibitors that were added to cultures at 12 h. For each inhibitor, the mean percentage±1 SEM of CD8+ T cell growth between 12 and 20 h relative to growth in the absence of inhibitor in the same experiment is shown (see Methods) (EIPA, $n = 8$; J/B, $n = 6$; PitStop, $n = 5$; LY294002, $n = 5$; EHT 1864, $n = 4$; IPA-3, $n = 4$; FTS, $n = 4$; Torin 1, $n = 5$ independent experiments). ** $P < 0.01$, *** $P < 0.001$, **** $P < 0.0001$ by Student's 1-sample 2-sided $t$-test. **b** Graph shows mean percentage macropinocytosis (Fig. 3b and Supplementary Fig. 2c) vs. mean percentage growth (**a**) for CD8+ T cells for each inhibitor. **c, d** Splenocytes (**c**) or purified pan T cells or CD4+ T cells (**d**) were stimulated with CD3/28 mAb for 12 h before incubation with DQ Red BSA for the indicated times at the indicated temperatures (**c**) or 4 h at 37 °C (**d**). Traffic of DQ Red BSA to lysosomes was assessed by flow cytometry (**c**) and confocal microscopy (**d**). Flow cytometry data are representative of four independent experiments (See Supplementary Fig. 3e). Arrows in (**d**) show accumulation of DQ Red BSA in LAMP-1-positive or LAMP-2-positive lysosomes as determined by staining with respective antibodies. The mean percentage ± 1 SEM of LAMP-1-positive or LAMP-2-positive lysosomes that contained DQ Red BSA per cell was calculated at 77.3 + 3.9 ($n = 35$ cells) and 84.9 + 3.8 ($n = 23$ cells), respectively. **e** Splenocytes were unstimulated or stimulated with CD3/28 mAb for different times in the absence or presence of inhibitors added at 12 h. Relative amounts of phospho-S6 in CD4+ T cells were determined by flow cytometry. All panels are from the same experiment. The red dashed line in each panel indicates the 12 h positive control. Data are representative of 6 independent experiments performed with EIPA and three independent experiments performed with J/B and PitStop (see Supplementary Fig. 3h). Source data are provided as Source Data file.

LY294002, EHT 1864 and IPA-3 also blocked mTORC1 activation (Supplementary Fig. 3h) and more potently than they inhibited macropinocytosis (Supplementary Fig. 2c), which could be explained by a role of the respective targets not only in macropinocytosis but also signaling pathways that regulate the Rheb small GTPase (Supplementary Fig. 6g)[25,26].

**mTORC1 activation by AA in T cells requires macropinocytosis.** To determine if macropinocytosis-acquired AA from the extracellular space directly for mTORC1 activation or indirectly as a consequence of protein uptake, we tested the effect of NH₄Cl upon sustained mTORC1 activation. NH₄Cl raises

intra-lysosomal pH and thereby inhibits protein degradation in lysosomes (Fig. 5a and Supplementary Fig. 6a, b). However, NH₄Cl had no effect upon sustained mTORC1 activation in CD4+ or CD8+ T cells when added to cultures at 12 h, thus showing that protein degradation is not necessary for this event (Fig. 5b and Supplementary Fig. 6c, d). To assess whether extracellular AA were sufficient for sustained mTORC1 activation, T cells were stimulated with CD3/28 mAb for 12 h in complete medium containing fetal calf serum (FCS), washed extensively and recultured for 2 h in minimal medium with or without AA. Reculture of T cells in media containing all 20 AA was sufficient to sustain activation of mTORC1 in CD4+ and CD8+ T cells when compared to T cells recultured in the absence of AA (Fig. 5c, d and

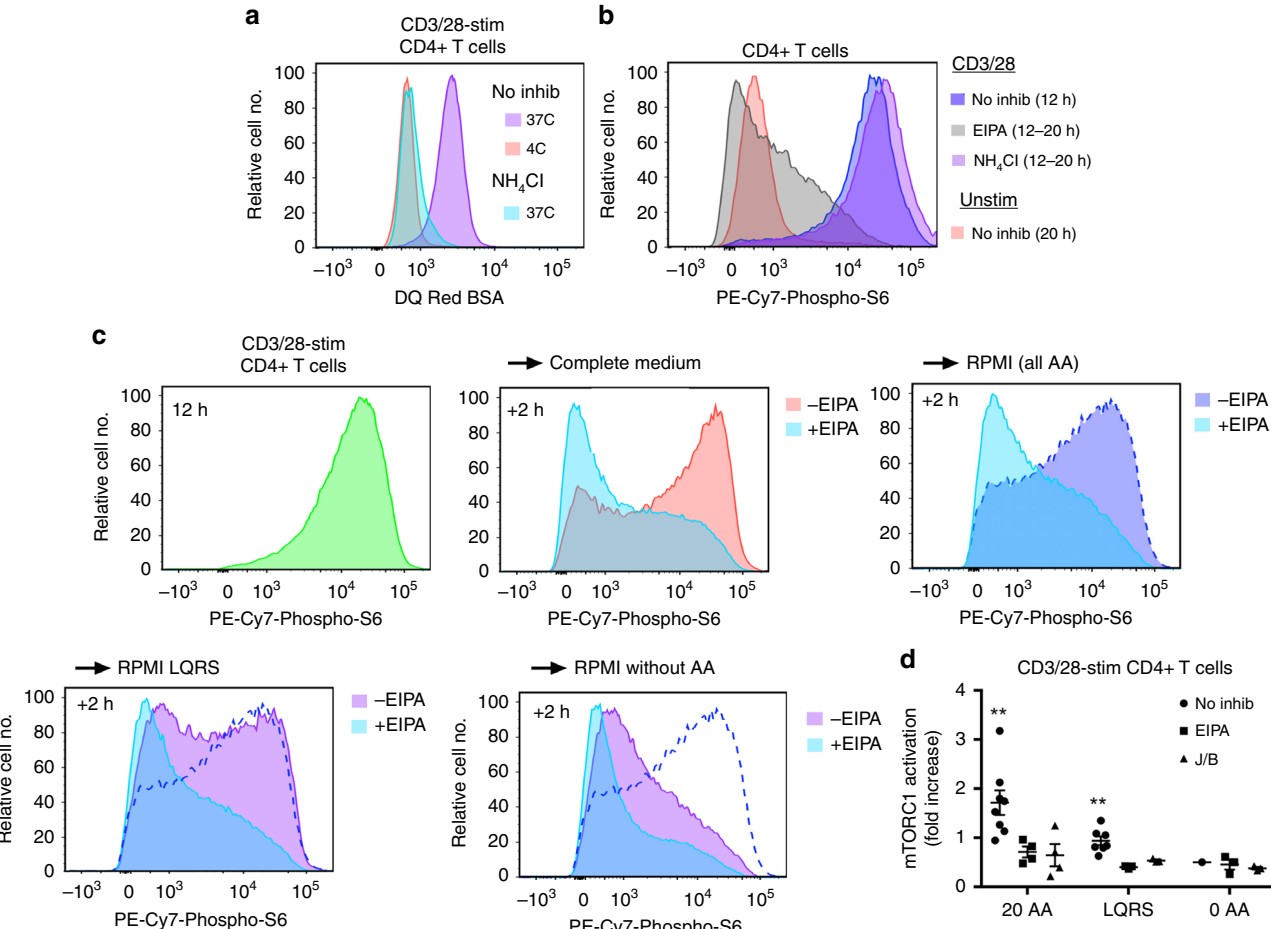

**Fig. 5 Macropinocytosis delivers free AA to T cells necessary for sustained activation of mTORC1. a, b** Splenocytes were stimulated with CD3/28 mAb for 12 h followed by incubation with DQ Red BSA for 4 h at the indicated temperatures in the presence or absence of $NH_4Cl$ (**a**) or for 8 h at 37 °C in the presence or absence of EIPA or $NH_4Cl$ (**b**). Flow cytometry histogram plots show DQ Red BSA fluorescence (**a**) or phospho-S6 levels (**b**) in CD4+ T cells. Plots are representative of three repeat experiments (see Supplementary Fig. 6b, d). **c** Splenocytes were stimulated with CD3/28 mAb in complete medium (RPMI plus FCS) for 12 h, washed and recultured in the indicated media for 2 h in the presence or absence of EIPA. Representative flow cytometry histograms show phospho-S6 levels in CD4+ T cells. All panels are from the same experiment. The mid-blue-dashed line indicates all AA in the absence of EIPA. **d** Mean±1 SEM of the fold increase in mTORC1 activation in CD4+ T cells at 14 h relative to the 0 AA control in the absence of inhibitors ($n = 8$ and 7 independent experiments for 20 AA and LQRS, respectively, in the absence of inhibitors; $n = 4$ for 20 AA and $n = 3$ independent experiments for LQRS and 0 AA in the presence of inhibitors). **P < 0.01 by Student's 1-sample 2-sided t-test. Source data are provided as Source Data file.

Supplementary Fig. 6e, f). Furthermore, re-culture of T cells in media containing only L, Q, R, and S AA, which have been implicated in lysosomal activation of mTORC1 previously[26], sustained mTORC1 activation compared to media that did not contain AA. Among these four AA, L and R are likely most important with regards mTORC1 activation. Thus, media that lack either L or R alone are unable to sustain mTORC1 activation whereas media that lack S or Q alone are able to sustain mTORC1 activation (Supplementary Fig. 7a, b), confirming and extending previous observations[29]. Significantly, sustained activation of mTORC1 by all AA or L, Q, R, and S was blocked by EIPA and J/B, thus indicating a requirement for macropinocytosis for AA-sustained mTORC1 activation (Fig. 5c, d and Supplementary Fig. 6e, f). Together, these data support a model in which macropinocytosis drives T cell growth through provision of free AA to an endolysosomal compartment necessary for sustained activation of mTORC1 (Supplementary Fig. 6g).

## Discussion

The findings reported herein constitute the first demonstration of a role for macropinocytosis in the growth of non-transformed

mammalian cells and suggest that this may be a more general phenomenon applicable to the growth of other primary cell types. Why T cells are dependent upon macropinocytosis for sustained mTORC1 activation and growth even under conditions where AA are freely available in the extracellular space remains to be determined. T cells express abundant AA transporters that could theoretically transport AA from the extracellular space to the cytosol and subsequently to lysosomes where they would promote mTORC1 activation. Two AA transporters, slc7a5 and slc1a5, involved in the transport of large neutral AA and glutamine, respectively, are required for mTORC1 activation in T cells[29,30]. However, it is unknown if these transporters are present on lysosomal membranes in T cells and if they are able to transport AA into this organelle. Instead, their role in mTORC1 activation could be consequent to their ability to transport AA from the extracellular space to the cytosol exclusively, at which site the imported AA would be detected by cytosolic AA sensors that are also known to control activation of mTORC1[26]. Potentially, activation of mTORC1 in T cells could be strictly dependent upon inputs from both cytosolic and lysosomal AA sensors. In this regard, a requirement for macropinocytosis for sustained mTORC1 activation in T cells could be explained by an

absence of relevant AA transporters at the lysosomal membrane necessary for import of AA from the cytosol into lysosomes and lysosomal sensing of AA (Supplementary Fig. 6g).

## Methods

**Animals**. Wild-type mice were bred in house and were on a mixed 129S6/SvEv X C57BL/6 genetic background. One exception was recipient mice in in vivo experiments that were on a CD45.1 C57BL/6 background (JAX). OTII TCR Tg mice (JAX) and *Rasgrp1* mutant mice (JAX) were on a C57BL/6 genetic background. Mice ranged in age from 6 weeks to 3 months. Mice of both sexes were used in experiments. All experiments performed with mice were in compliance with University of Michigan guidelines and were approved by the University Committee on the Use and Care of Animals.

**T cell macropinocytosis**. Murine splenocytes from wild-type or *Rasgrp1* mutant mice or pan T cells, purified from splenocytes of wild-type mice by column depletion (Miltenyi Biotec), were resuspended in RPMI 1640 medium (Thermo Fisher) supplemented with 10% heat-inactivated FCS (Gibco). Splenocytes were seeded into U-bottomed 96-well plates at a density of $1 \times 10^6$ cells per well and were stimulated or not with anti-CD3 (1 µg/ml; eBioscience, clone 145–2C11) and anti-CD28 (1 µg/ml; eBioscience, clone 37.51) mAb for the indicated times. Pan T cells were seeded at a density of $1 \times 10^6$ cells per well into the wells of flat-bottomed 96-well plates pre-coated with anti-CD3 mAb (10 µg/ml) and soluble CD28 mAb (1 µg/ml) was added to wells. 70 kDa Fdex, BSA-Alexa 488, or DQ Red BSA (all Thermo Fisher) macropinocytosis probes were added to wells at final concentrations of 1 mg/ml, 0.4 mg/ml, and 50 µg/ml, respectively, at the indicated times. Incubation with probes was for the indicated times at 37 °C or 4 °C. Pharmacological inhibitors were added to cultures 15 min prior to addition of macropinocytosis probes in a range of concentrations as indicated or at the following final concentrations: EIPA (Sigma), 50 µM; jasplakinolide (Tocris), 1 µM; (S)-(-)-blebbistatin (Tocris), 75 µM; PitStop 2 (Sigma), 25 µM; FTS (Sigma), 25 µM; LY294002 (Cayman), 50 µM; EHT 1864 (Cayman), 10 µM; IPA-3 (Tocris), 20 µM; Torin 1 (Tocris), 500 nM; NH$_4$Cl (Sigma), 10 mM. Cells were harvested, washed, stained with APC-Cy7-CD4 (BD Pharmingen, clone GK1.5, cat. no. 552051, dilution 1:100) and APC-CD8α (BD Pharmingen, clone 53-6.7, cat. no. 553035, dilution 1:100) mAb and analyzed by flow cytometry on BD Fortessa or BD FACSCanto instruments (BD Biosciences). Gating strategies are illustrated in Supplementary Fig. 8. Percentage macropinocytosis in the presence of inhibitors was calculated as follows: [(MFI in presence of inhibitor at 37 °C−MFI in absence of inhibitor at 4 °C)/(MFI in absence of inhibitor at 37 °C−MFI in absence of inhibitor at 4 °C)] × 100. Percentage inhibition of DQ Red BSA fluorescence in the presence of NH$_4$Cl was calculated as follows: [(MFI in absence of inhibitor at 37 °C−MFI in presence of NH$_4$Cl at 37 °C)/(MFI in absence of inhibitor at 37 °C−MFI in absence of inhibitor at 4 °C)] × 100.

To assess human T cell macropinocytosis, human peripheral blood mononuclear cells (PBMC) were isolated from buffy coats obtained from the New York Blood Center and resuspended in RPMI 1640 with 10% FCS. PBMC were seeded into 96 well U-bottomed plates at a density of $5 \times 10^5$ cells per well and were stimulated or not with anti-CD3 (1 µg/ml; Invitrogen, clone OKT3) and anti-CD28 (1 µg/ml; Invitrogen, clone CD28.2) mAb or PHA (1.5% final; Thermo Fisher) for 20 h. Cells were incubated with BSA-Alexa 488 at 0.4 mg/ml for the last 8 h of culture. EIPA and J/B were added to cultures 15 min prior to addition of probe at the above concentrations. Cells were harvested, stained with APC-Cy7-CD4 (Biolegend, clone RPA-T4, cat. no. 300518, dilution 1:100) or PerCP-Cy5-5-A-CD4 (Biolegend, clone OKT4, cat. no. 317428, dilution 1:100) and Alexa 700-CD8α (Biolegend, clone SK1, cat. no. 344724, dilution 1:100) or BV-605-CD8 (Biolegend, clone RPA-T8, cat. no. 301040, dilution 1:20) mAb and analyzed by flow cytometry. The gating strategy is illustrated in Supplementary Fig. 8.

**T cell growth**. Murine splenocytes were stimulated with CD3/CD28 mAb as above at 37 °C for 12 or 20 h in the presence or absence of inhibitors that were added at 12 h. Cells were harvested, washed, stained with APC-Cy7-CD4 and APC-CD8α mAb and analyzed by flow cytometry. Median FSC-A of CD4+ and CD8+ T cells was taken as a relative measure of cell size. In each experiment, the effect of inhibitors upon T cell growth between 12 and 20 h was calculated as a percentage of T cell growth observed in the absence of inhibitor as follows: [(FSC-A in presence of inhibitor at 20 h−FSC-A in absence of inhibitor at 12 h)/(FSC-A in absence of inhibitor at 20 h−FSC-A in absence of inhibitor at 12 h)] × 100.

**Macropinocytosis in vivo**. One million TCR Vβ5+ CD4+ T cells from CD45.2 OTII TCR Tg mice were injected into the tail veins of CD45.1 wild-type recipients. After 24 h, recipient mice were immunized i.d. in footpads with Ova (0.5 mg/ml) in RIBI adjuvant (Sigma) or with RIBI adjuvant alone (25 µl per footpad). Twelve hours later, all mice were injected i.d. in footpads with BSA-Alexa 647 (5 mg/ml in 25 µl of PBS per footpad; Thermo Fisher) and mice were euthanized after an additional 8 h. Draining popliteal lymph nodes were harvested from mice, stained with PerCP-Cy5-5-A-CD45.2 (BioLegend, clone 104, cat. no. 109828, dilution 1:200), Alexa Fluor 700A-CD45.1 (BioLegend, clone A20, cat. no. 110724, dilution 1:200), V500-CD4 (BD BioSciences, clone RM4-5, cat. no. 560782, dilution 1:400),

APC-Cy7- CD8α (Invitrogen, clone 53-6.7, cat. no. A15386, dilution 1:100), and PE-TCR V-beta5 (BD Pharmingen, clone MR9-4, cat. no. 553190, dilution 1:200) mAb, and analyzed by flow cytometry to assess BSA-Alexa 647 uptake by OTII TCR Tg T cells. The gating strategy is illustrated in Supplementary Fig. 8.

**Confocal microscopy**. Murine splenic pan T cells or CD4+ T cells (isolated by column depletion) were stimulated with CD3 and CD28 mAb for 12 h as above before incubation in the presence or absence of BSA-Alexa 488 (0.4 mg/ml) or DQ Red BSA (50 µg/ml) at 37 °C or 4 °C in the presence or absence of EIPA (50 µM) for a further 4 or 8 h. Cells were harvested, washed, resuspended in PBS and sedimented for 1 h on ice at 1 g onto coverslips previously coated with 0.1% poly-L-lysine (Sigma). Cells were fixed in situ by addition of an equal volume of 4% paraformaldehyde and incubation for 30 min at room temperature. Coverslips were then washed and cells were stained with rat anti-mouse CD4 (R&D Systems, clone GK1.5), CD8α (R&D Systems, clone 53-6.7), LAMP-1 (eBioscience, clone 1D4B, cat. no. 14-1071-82, dilution 1:100), or LAMP-2 (Invitrogen, clone M3/84, cat. no. MA5–17861, dilution 1:100) mAb overnight at 4 °C. The following day, coverslips were washed, blocked with 5% donkey serum for 1 h, incubated with Alexa 488- or Alexa 594-labeled donkey anti-rat secondary antibody (Thermo Fisher, cat. nos. A-21208 and A-21209, dilution 1:200), and stained with 10 µg/ml Hoechst 33258 (Thermo Fisher). For imaging of polymerized filamentous actin, cells were permeabilized with Triton X-100 and stained with Acti-stain-488 fluorescent phalloidin (0.1 µM; Cytoskeleton, Inc.) prior to the antibody staining steps. Coverslips were mounted in ProLong Gold anti-fade (Thermo Fisher) and images were captured on a Leica upright SP5 confocal microscope (Leica). Data showing the percentage of T cells with indicated numbers of macropinosomes are based upon counts of >150 cells.

**Scanning EM**. Murine BM-derived macrophages were prepared as described[31]. After 6 d culture, macrophages were stimulated for 15 min with CSF1. Murine CD4+ T cells were stimulated for 16 h with CD3/28 mAb and prepared as indicated in confocal microscopy.

Macrophages and T cells were processed for scanning EM as described[32] except that the fixed samples were dehydrated in an acetone series followed by hexamethyldisalizane. Air-dried samples were gold coated and imaged on an Amray 1900 field emission scanning electron microscope.

**Flow cytometric analysis of mTORC1 and NFκB activation**. Murine splenocytes were stimulated as above for the indicated times in the presence or absence of inhibitors at concentrations indicated above. Cells were harvested, fixed in 4% paraformaldehyde for 20 min at room temperature, washed with PBS supplemented with 5% FCS, and permeabilized by drop-wise addition of ice-cold, 90% methanol with gentle vortexing. Cells were washed and stained with APC-Cy7-CD4, APC-CD8α and PE-Cy7-phospho-S6 mAb (Cell Signaling Technology, clone D57.2.2E, cat. no. 34411, dilution 1:50) or anti-phospho-NFκB p65 (pS356) (Cell Signaling Technology, clone 93H1, cat. no. 3033, dilution 1:100) followed by Alexa488-labeled donkey anti-rabbit secondary antibody (Jackson Immunoresearch, cat. no. 711-545-152, dilution 1:100) and analyzed by flow cytometry. The gating strategy is illustrated in Supplementary Fig. 8. In each experiment, the effect of inhibitors upon mTORC1 activation between 12 and 20 h or NFκB activation between 12 and 14 h was calculated as a percentage of mTORC1 or NFκB activation observed in the absence of inhibitor as follows: [(MFI phospho-S6 or NFκB of CD3/28 stimulated T cells in presence of inhibitor−MFI phospho-S6 or NFκB of unstimulated T cells in absence of inhibitor)/(MFI phospho-S6 or NFκB of CD3/28 stimulated T cells in absence of inhibitor−MFI phospho-S6 or NFκB of unstimulated T cells in absence of inhibitor)] × 100.

For experiments that determined if AA were sufficient to sustain mTORC1 activation, splenocytes were stimulated for 12 h with CD3/28 mAb in RPMI 1640 with 10% FCS as above. Cells were then washed extensively in PBS and recultured in wells of 96 well U-bottomed plates at $1 \times 10^6$ cells per well in RPMI 1640 with 10% FCS, or with modified RPMI 1640 without AA (USBiological) to which different combinations of AA (Thermo Fisher or Sigma) had been added at the concentrations normally found in RPMI 1640. EIPA and J/B were added to some wells at the same concentrations indicated above. After 2 h, cells were harvested and analyzed for phospho-S6 by flow cytometry.

**Western blotting**. Purified splenic CD4+ T cells were stimulated with Mouse T-Activator CD3/CD28 Dynabeads (Life Technologies) for the indicated times before cell disruption by boiling in reducing SDS-PAGE sample buffer. Activation of mTORC1, ERK MAPK and NFκB was determined by Western blotting using anti-phospho-S6 (Cell Signaling Technology, cat. no. 2211, dilution 1:1000), anti-p44/p42 MAPK (T202/Y204) (Cell Signaling Technology, clone E10, cat. no. 9106, dilution 1:1000), or anti-phospho IκBa (S32/36) (Cell Signaling Technology, clone 5A5, cat. no. 9246, dilution 1:1000) antibodies, respectively. Blots were stripped and reprobed with antibodies against unphosphorylated MAPK (Cell Signaling Technology, clone 137F5, cat. no. 137F5, dilution 1:1000), S6 (Cell Signaling Technology, clone 54D2, cat. no. 2317, dilution 1:1000), or IκBa (Cell Signaling Technology, clone no. 9242, dilution 1:1000).

**Statistical analysis**. *P* values were calculated using Student's 1-sample or 2-sample 2-sided *t*-tests as appropriate for normally distributed data.

**Reporting summary**. Further information on research design is available in the Nature Research Reporting Summary linked to this article.

## Data availability

The flow cytometry and imaging data supporting these findings are available from the corresponding author upon request. The source data underlying Fig. 1c, h, k, n, 3d, f, h, 4a, 5e, and Supplementary Figs. 1e, g, 2a–c, e–g, 3b, e, h, 4a–d, f, 5b–e, 6b, d, f, and 7b is provided as a Source Data File.

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

## Acknowledgements

J.C.C. was supported by NIH AI T32007413 Research Training in Experimental Immunology Training Grant and a Rackham Graduate School Merit Fellowship. J.A.S. was supported by NIH grant R01 GM110215.

## Author contributions

J.C.C., J.S., I.G. and P.D.K. designed experiments and analyzed data; J.C.C. conducted experiments with assistance from D.C., P.E.L., J.T. and J.S.; P.D.K. wrote the manuscript with input from J.C.C. and J.S.; P.D.K. supervised the project.

## Competing interests

The authors declare no competing interests.
