## [Peer Review File · Nature Communications]

Reviewers' comments:

Reviewer #1 (Macropinocytosis, cell biology)(Remarks to the Author):

The manuscript by Charpentier et al. "Macropinocytosis drives T cell growth by sustaining the activation of mTORC1", describes the role of macropinocytosis in the growth of T cells (CD4 and CD8) and the role of the activation of mTORC1 in this process.

The group use mainly cytofluorometry and some confocal microscopy in the study, they also performed some macropinocytosis studies in vivo (mouse model) and few approaches with peripheral human T cells.

Their results are based on the activation of T cells mainly with CD3/28 monoclonal antibodies, and to determine that T cells are performing macropinocytosis they use two fluid phase markers: 70 kDa fluorescein-dextran and BSA-fluorochrome labeled. Their study in many aspects is based on the use of inhibitors.

Beside the fact that macropinocytosis is involved in the growth of primary T cells (CD4 or CD8), other interesting contribution of the study is the fact that mTORC1 activation is involved in the growth of the T cells, and that macropinocytosis also contribute to T cell growth through provision of free amino acids to an endolysosomal compartment necessary for sustained activation of mTORC1.

The manuscript is well written and in spite of so much work done, they present the most concrete data that supports their conclusions. This study points out the relevance of macropinocytosis in the growth of primary cells, but also has an impact on immunology, since the description of the study falls into two of the most important immune cell types: CD4 T and CD8 T lymphocytes.

However I have some concerns:

1.- Most of the study is performed on mouse T cells, and some few experiments are performed on primary peripheral human T cells, however the conclusions are generalized for both species. Beside there are differences in the activation of murine and human cells (CD3/CD28 mAb and PHA respectively), so conclusions should not be overestimated for human T cells based on murine cells experiments.

2.- In the figure 1g (T cells activated with CD3/CD28 and incubated at 37°C, page 29), I have a concern with the distribution of the CD4 label (red) which is observed homogeneously around the cells where the label is observed. I wonder if during the macropinocytosis process, there are important cytoskeleton rearrangements that driven ruffling and macropinosome formation, so ¿ it should not be expected an alteration in the CD4 label distribution as well as in cell morphology of activated T cells? . The same observation is valid for figure 2c.

3.- Considering that the authors conclusion that macropinocytosis drives T cell growth, it will be interesting to have images of cytoskeleton reorganization during macropinocytosis of T cells, which the authors probed that is involved, since they use cytoskeleton inhibitors that had a profound effect on fluid phase uptake (results in figures 2 and 3 and their corresponding extended data figures). Beside one of the hallmarks of macropinocytosis are the formation of membrane protrusions and the formation of the macropinocytic cup, and since the authors' claim that "T cell macropinocytosis differs from classical macropinocytosis in that it is independent of Ras, although shows partial dependency upon PI3K, Rac1 and Pak1", it will be very interesting to visualize the characteristics of these events (cytoskeleton and plasma membrane rearrangements during macropinocytosis) in the activated T cells

(pan T , T CD4 or T CD8), either by confocal microscopy or better by scanning electron microscopy.

I consider that this manuscript describes an original approach to study macropinocytosis in primary cell types and will contribute to the understanding of this unique cell process that is not yet fully understood.

Reviewer #2 (T cell metabolism, T cell activation)(Remarks to the Author):

Charpentier et al showed an important role of micropinocytosis in T cell growth and mTORC1 activation. Overall, the study is well designed and the results are largely convincing, but some important concerns need to be addressed.

1. Clearly, inhibitors of micropinocytosis have very potent effects on T cell growth and mTORC1 activation. Given such potent effects, the authors need to include a careful dose response curve to show their effects on micropinocytosis, T cell growth and mTORC1 activity.

2. The authors ascribe the effects of micropinocytosis on T cell growth to mTORC1 activation. Is the effect specific to mTORC1, or does micropinocytosis inhibition have a general effect on T cell signaling? It is therefore important for the authors to examine other TCR signaling pathways, e.g. MAPK, NFkB and NFAT.

3. The authors showed that inhibitors of PI3K, Rac1 or Pak1 affected micropinocytosis and T cell growth. Do they also affect mTORC1 activity?

4. Does micropinocytosis affect T cell growth and mTORC1 activity in vivo? There are no functional data in vivo.

5. The authors added a combination of amino acids LQRS to T cells for mTORC1 activation. Why? Can they show the data from individual amino acids of these 4 amino acids, and a combination of them?

6. The lysosomal staining data in Fig. 3d is not very compelling. There are clearly large lysosomes that are unlabeled. Also, there are a number of LAMP-1+ signals that are smaller in size – are these also lysosomes? Can the authors use dyes other than LAMP-1 to label lysosomes?

Reviewer 1

1.- Most of the study is performed on mouse T cells, and some few experiments are performed on primary peripheral human T cells, however the conclusions are generalized for both species. Beside there are differences in the activation of murine and human cells (CD3/CD28 mAb and PHA respectively), so conclusions should not be overestimated for human T cells based on murine cells experiments.

Thank you for this comment. In response, we have conducted additional experiments with human T cells. Please see new Fig. 1l-n and Fig. 3e,f, which show CD3/28 mAb-stimulated uptake of macropinocytosis probe and its inhibition by EIPA and J/B.

2.- In the figure 1g (T cells activated with CD3/CD28 and incubated at 37°C, page 29), I have a concern with the distribution of the CD4 label (red) which is observed homogeneously around the cells where the label is observed. I wonder if during the macropinocytosis process, there are important cytoskeleton rearrangements that driven ruffling and macropinosome formation, so ¿ it should not be expected an alteration in the CD4 label distribution as well as in cell morphology of activated T cells? . The same observation is valid for figure 2c.

In Fig. 2d, we provide new confocal images that capture polymerized F-actin in a forming macropinocytic cup of a CD3/28-stimulated T cell. CD4 is not displaced from the forming cup but is instead enriched in this structure. We hope that this addresses the reviewer's concern.

3.- Considering that the authors conclusion that macropinocytosis drives T cell growth, it will be interesting to have images of cytoskeleton reorganization during macropinocytosis of T cells, which the authors probed that is involved, since they use cytoskeleton inhibitors that had a profound effect on fluid phase uptake (results in figures 2 and 3 and their corresponding extended data figures). Beside one of the hallmarks of macropinocytosis are the formation of membrane protrusions and the formation of the macropinocytic cup, and since the authors' claim that "T cell macropinocytosis differs from classical macropinocytosis in that it is independent of Ras, although shows partial dependency upon PI3K, Rac1 and Pak1", it will be very interesting to visualize the characteristics of these events (cytoskeleton and plasma membrane rearrangements during macropinocytosis) in the activated T cells (pan T , T CD4 or T CD8), either by confocal microscopy or better by scanning electron microscopy.

A very important point. In response, in Fig. 2a,b we now provide scanning EM images of developing macropinosomes in unstimulated and CD3/28-stimulated T cells that show macropinocytic cups in various stages of formation. For comparison,

we also provide scanning EM images of macrophages stimulated with CSF1. The structures in the two cell types are identical in appearance.

Reviewer 2.

1. Clearly, inhibitors of micropinocytosis have very potent effects on T cell growth and mTORC1 activation. Given such potent effects, the authors need to include a careful dose response curve to show their effects on micropinocytosis, T cell growth and mTORC1 activity.

For the two potent inhibitors, EIPA and J/B, this information is now provided in Fig. 3c,d (macropinocytosis), Extended Data Fig. 4a,b (growth), and Extended Data Fig. 4c,d (mTORC1 activation). Both inhibitors block all three types of response at concentrations that have been previously recognized to inhibit macropinocytosis. At each tested drug concentration, inhibitory effects upon each response are broadly similar.

2. The authors ascribe the effects of micropinocytosis on T cell growth to mTORC1 activation. Is the effect specific to mTORC1, or does micropinocytosis inhibition have a general effect on T cell signaling? It is therefore important for the authors to examine other TCR signaling pathways, e.g. MAPK, NFkB and NFAT.

This is a good question, which we have explored. We focused upon NFkB and MAPK pathways since phospho-specific reagents are available that allow ready detection of activation by flow cytometry. Inhibition of macropinocytosis by EIPA does not impair NFkB activation, whereas inhibition of macropinocytosis by J/B increases NFkB activation in the 12-14 h period. This data is now provided in Extended Data Fig. 4e-h. This is in stark contrast to the effect of EIPA and J/B upon mTORC1 activation between 12-14 h (compare Fig.4e). Thus, the inhibitory effect of effect of EIPA and J/B upon mTORC1 appears to be specific and not a consequence of a general interruption of T cell signaling. Activation of MAPK between 12 and 20 h is marginal even in the absence of inhibitor and thus it has been more difficult to assess if inhibitors influence this pathway.

3. The authors showed that inhibitors of PI3K, Rac1 or Pak1 affected micropinocytosis and T cell growth. Do they also affect mTORC1 activity?

Yes they do and this information is now included in Extended Data Fig.3h. An associated narrative that explains these observations is provided on page 8 of the text.

4. Does micropinocytosis affect T cell growth and mTORC1 activity in vivo? There are no functional data in vivo.

Unfortunately, this point is impossible to address currently as EIPA and J/B show significant toxicity in vivo. In addition, EIPA has poor solubility and precipitates out of solution at the injection site. Chemical modification of EIPA will be necessary to overcome the solubility problem in vivo.

5. The authors added a combination of amino acids LQRS to T cells for mTORC1 activation. Why? Can they show the data from individual amino acids of these 4 amino acids, and a combination of them?

LQRS were chosen because these amino acids are recognized activators of mTORC1 (reviewed in ref. 26). Individually, each amino acid is not sufficient to sustain mTORC1 activation. Nonetheless, we provide data to show that L and R are probably more significant than Q and S. This can be concluded from the observation that media that do not contain L or R are unable to sustain mTORC1 activation whereas media that do not contain Q or S are able to sustain mTORC1 activation. This new data is provided in Extended Fig.6a,b.

6. The lysosomal staining data in Fig. 3d is not very compelling. There are clearly large lysosomes that are unlabeled. Also, there are a number of LAMP-1+ signals that are smaller in size – are these also lysosomes? Can the authors use dyes other than LAMP-1 to label lysosomes?

We have enlarged the panel concerned to better illustrate this point (see new Fig. 4d). The presence of red fluorescent DQRedBSA signals within LAMP-1 positive lysosomes is unequivocal. Note that red fluorescence is only possible upon lysosomal delivery. Note also that flow cytometry data provided in Fig.4c and Extended Data Fig.3d,e also reveals lysosomal targeting. The reviewer is correct that some LAMP1+ lysosomes in Fig.4d do not contain DQRedBSA. Presumably this is because macropinosomes containing DQRedBSA were not delivered to those particular lysosomes.

Reviewers' comments:

Reviewer #1 (Remarks to the Author):

The manuscript by Charpentier et al. "Macropinocytosis drives T cell growth by sustaining the activation of mTORC1", describes the role of macropinocytosis in the growth of T cells (CD4 and CD8) and the role of the activation of mTORC1 in this process.

The group used mainly cytofluorometry and some confocal microscopy in the study, and in the revised version, they included some scanning electron microscopy. Most of the strategies were based on the use of inhibitors.

In the revised version, the authors included a new set of experiments, or extended the previous observations, making the new version a more robust and conclusive version of the manuscript.

I consider that the inclusion of the new set of experiments answered adequately my original observations and as I mentioned before, improved the manuscript substantially.

Reviewer #2 (Remarks to the Author):

The authors have address many of my previous comments, but some important questions still remain. For the original question #2, the authors analyzed NFkB and MAPK activities at 14 hours after stimulation, but these are not the proper time points. It is well known that TCR stimulation results in rapid activation of NFkB, MAPK and NFAT activities within minutes, and the standard measurement is western blot, not flow cytometry. The authors need to use the proper time points and measurements, and to exclude the global and non-specific effects of these drugs on these other ways. In fact, based on the upregulation of NFkB activity in drug-treated cells, the drug off-target effect remains a concern, and thus proper doses should be tested.

For my question #6, the authors failed to address it. They need to use additional dyes other than LAMP-1 to label lysosomes, as what was requested previously, and also include statistical analysis to support their conclusions. In addition, the authors indicated the use of flow cytometry method for lysosomal targeting (Fig. 4c), but how reliable and specific is this approach? The authors need to rigorously validate the flow cytometry approach, and also provide statistical analysis.

Reviewer #1 (Remarks to the Author):

In the revised version, the authors included a new set of experiments, or extended the previous observations, making the new version a more robust and conclusive version of the manuscript. I consider that the inclusion of the new set of experiments answered adequately my original observations and as I mentioned before, improved the manuscript substantially.

We thank the reviewer for these comments.

Reviewer #2 (Remarks to the Author):

1. The authors have address many of my previous comments, but some important questions still remain. For the original question #2, the authors analyzed NFkB and MAPK activities at 14 hours after stimulation, but these are not the proper time points. It is well known that TCR stimulation results in rapid activation of NFkB, MAPK and NFAT activities within minutes, and the standard measurement is western blot, not flow cytometry. The authors need to use the proper time points and measurements, and to exclude the global and non-specific effects of these drugs on these other ways.

Thank you for these comments. We have now assessed the effect of EIPA (the gold standard macropinocytosis inhibitor) upon mTORC1, MAPK and NFkB in the acute T cell activation setting using Western blotting. The results are shown in new Extended Figure 5. EIPA inhibits the activation of mTORC1 but not MAPK or NFkB.

2. For my question #6, the authors failed to address it. They need to use additional dyes other than LAMP-1 to label lysosomes, as what was requested previously, and also include statistical analysis to support their conclusions. In addition, the authors indicated the use of flow cytometry method for lysosomal targeting (Fig. 4c), but how reliable and specific is this approach? The authors need to rigorously validate the flow cytometry approach, and also provide statistical analysis.

Thank you for this comment. We have now conducted additional co-staining with LAMP-2 antibodies and the same results are observed and data is shown in an updated Fig. 4. A quantification of the percentage of LAMP-1+ and LAMP-2+ lysosomes per cell that contain DQ Red BSA is included in the legend to the figure.

DQ Red BSA is highly reliable indicator of lysosomal targeting. It is a modified form of BSA that is heavily labeled with BODIPY Texas-Red-X dye. The high density of labeling results in complete quenching of fluorescence so that, when

excited with 590 nm wavelength light, 620 nm emission fluorescent light is not emitted from the molecule. However, upon proteolytic digestion of the BSA, the quenching effect is relieved and the generated peptide fragments readily emit 620 nm wavelength light when excited with 590 nm light. Critically, DQ-BSA can only emit red fluorescence if it is delivered to lysosomes that are the only organelle within the cell where proteolytic digestion of ingested probe could take place.

In Figure 4c and Extended Data Figure 3d the acquisition of red fluorescence by CD4 and CD8 T cells cultured with DQ-Red BSA at 37C but not 4C is shown. Experiments were repeated 4 times and the summary statistics and statistical analyses are provided in Extended Data Figure 3e. We also provide data in Figure 5a and Extended Data Figure 6a that shows that raising lysosomal pH, which blocks proteolytic digestion in the lysosome, completely prevents the acquisition of red fluorescence by CD4 and CD8 T cells cultured with DQ-Red BSA. Experiments were repeated 3 times and statistical analysis of this effect is provided in Extended Data Figure 6b.

A large number of studies have used DQ-labeled BSA to monitor lysosomal targeting of BSA previously. Following are some select references:

1. Commisso C, Davidson SM, Soydaner-Azeloglu RG, Parker SJ, Kamphorst JJ, Hackett S, Grabocka E, Nofal M, Drebin JA, Thompson CB, Rabinowitz JD, Metallo CM, Vander Heiden MG, Bar-Sagi D. Macropinocytosis of protein is an amino acid supply route in Ras-transformed cells. *Nature*. 2013 May 30;497(7451):633-7.
2. Marwaha R, Sharma M. DQ-Red BSA Trafficking Assay in Cultured Cells to Assess Cargo Delivery to Lysosomes. *Bio Protoc*. 2017 Oct 5;7(19).
3. Ahram M, Sameni M, Qiu RG, Linebaugh B, Kirn D, Sloane BF. Rac1-induced endocytosis is associated with intracellular proteolysis during migration through a three-dimensional matrix. *Exp Cell Res*. 2000 Nov 1;260(2):292-303.
4. Reis RC, Sorgine MH, Coelho-Sampaio T. A novel methodology for the investigation of intracellular proteolytic processing in intact cells. *Eur J Cell Biol*. 1998 Feb;75(2):192-7.

REVIEWERS' COMMENTS:

Reviewer #2 (Remarks to the Author):

The authors have successfully addressed my previous critiques.